# Exploiting negative photochromism to harness a four-photon-like fluorescence response with two-photon excitation

Carlos Benitez-Martin [1,2], Jean Rouillon[1], Eduard Fron[3,4], Flip de Jong [3,4], Morten Grøtli [2] ✉, Johan Hofkens [3,4] ✉, Uwe Pischel [5] ✉ & Joakim Andréasson [1] ✉

Combining nonlinear optical processes and photoswitching transcends the limitations of conventional and even standalone super-resolution imaging. While photoswitching enables resolution improvement, it is typically constrained by limited imaging depth, potential phototoxicity and the low number of inherently fluorescent photoswitches. Nonlinear excitation, such as two-photon absorption, addresses some of these challenges. Here, we present a molecular design strategy that unites the molecular control of T-type negative photoswitches (PS) and two-photon absorption. In these designs, two-photon absorbing push-pull fluorophores that function as FRET-donors are linked to T-type negative PS FRET-acceptors, e.g., donor-acceptor Stenhouse adducts (DASA) or 1,1'-binaphthyl-bridged imidazole dimers. FRET-sensitized isomerization of PS is delicately balanced by reverse thermal isomerization and results in nonlinearly potentiated fluorescence with a quartic fluorescence response upon two-photon excitation, implying enhanced spatial resolution potential. The use of T-type PS is instrumental to this approach, as it ensures temporally stable photonic responses and recyclability without incurring irreversible saturation effects.

Multiphoton processes are crucial in various scientific and technological fields due to their unique ability to interact with matter in ways that single-photon processes cannot. A prime example is multiphoton microscopy, which enables high-resolution imaging with deeper tissue penetration and reduced photodamage[1,2]. The utility of multiphoton processes is founded on the nonlinear dependence on the excitation probability as a function of the intensity of the excitation light. For instance, in two-photon absorption (2PA) a quadratic dependence applies[3–6], while for higher-order multiphoton absorption, cubic (3PA)[7] or quartic (4PA) dependencies are observed[8]. This implies that the

larger the number of photons of the same energy being coherently absorbed in the excitation process, the more tightly the localization of the excited chromophores is around the focal point. When the subsequently emitted fluorescence is monitored in fluorescence microscopy, it follows that the 3D resolution increases accordingly. Thus, compared to two-photon microscopy, three- and four-photon imaging[9–12] offer improved spatial resolution at a given excitation wavelength. However, three- and four-photon techniques are bound to significant barriers, including very high excitation intensities and limited availability of specifically designed fluorophores and optical

[1]Department of Chemistry and Chemical Engineering, Physical Chemistry, Chalmers University of Technology, Göteborg, Sweden. [2]Department of Chemistry and Molecular Biology, University of Gothenburg, Göteborg, Sweden. [3]Chem&Tech – Molecular Imaging and Photonics, KU Leuven, Celestijnenlaan 200F, Leuven, Belgium. [4]KU Leuven Core Facility for Advanced Spectroscopy, Celestijnenlaan 200F, Leuven, Belgium. [5]CIQSO – Center for Research in Sustainable Chemistry and Department of Chemistry, University of Huelva, Campus de El Carmen s/n, Huelva, Spain. ✉e-mail: grotli@chem.gu.se; johan.hofkens@kuleuven.be; uwe.pischel@diq.uhu.es; a-son@chalmers.se

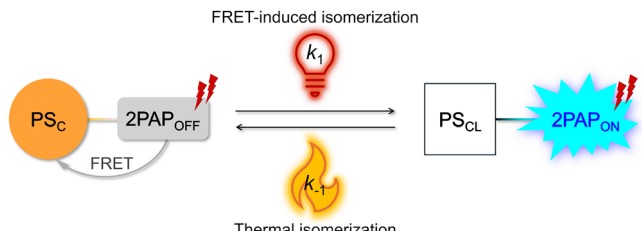

**Fig. 1 | Representation of the overall 4for2 function. In the examples developed in this study, the two-photon excitation process corresponds to the use of 800-nm photons.** $PS_C$: Colored FRET-active form of the photoswitch. $PS_{CL}$: Colorless form of the photoswitch. 2PAP: Two-photon absorbing probe.

systems. As a result, their application remains of merely academic interest.

In parallel, photoswitching has emerged as a cornerstone principle[13] in the pursuit of imaging beyond the diffraction limit, which historically constrained optical resolution to ~200 nm laterally and ~500 nm axially. In super-resolution microscopy methods such as stochastic optical reconstruction microscopy (STORM)[14,15], photo-activated localization microscopy (PALM)[16,17], and reversible saturable optical linear fluorescence transitions (RESOLFT)[18], photoswitchable fluorophores[19–21] are critical for achieving nanoscale resolution.

The synergistic integration of nonlinear excitation and photoswitching will unlock opportunities for high-resolution imaging in challenging contexts. By combining the depth capabilities of nonlinear processes with the molecular precision of photoswitching, advanced hybrid modalities have the potential to empower researchers to explore (biological) structures and functions with improved spatial and temporal resolution.

In this context, it would be a major step forward if 2PA molecular systems could be designed in conjunction with photoswitches so that they mimic the photophysical behavior of higher-order photon-absorbing systems. The herein presented work addresses bespoke challenges[19] and shows that the integration of photoswitchable molecules with fluorescent probes allows for a four-photon-like fluorescence response, albeit only two-photon excitation was used[22–24]. Our 4for2 approach relies on the combination of two sequential 2PA processes (two-photon excited fluorescence and 2PA-induced FRET-sensitized photoswitching)[22] in conjunction with T-type negative photoswitches (PS)[25,26]. This design produces a quartic (4PA) dependence of fluorescence on the excitation light intensity, implying an improved resolution potential compared to conventional 2PA fluorescent probes. Furthermore, it can be combined with on/off switching of subsets of the PS conjugates to potentially realize further resolution improvements. In here, we demonstrate the key features of our concept in solution, which provides a controlled environment to rigorously validate the underlying photophysical mechanisms, free from the potential complications of heterogeneous or cellular systems. This proof-of-principle study allows an unambiguous demonstration of the concatenated two-photon processes and photoswitching reversibility, laying a robust foundation for subsequent implementation in more complex environments such as solid-state matrices or biological tissues. Furthermore, it enables the precise characterization of excitation-intensity dependencies and photoswitch fatigue resistance, which are critical parameters for future translation into nonlinear super-resolution microscopy platforms. Crucial to our approach is the reversibility of photoswitching, which guarantees a low and controlled steady-state concentration of the emissive species. In this context, it is noteworthy that the use of negative T-type PS enables the use of a single operation wavelength, accessible to a large majority of 2PA chromophores (e.g., 800 nm) and dispenses technologically more complex and costly two-color laser experiments[23,24].

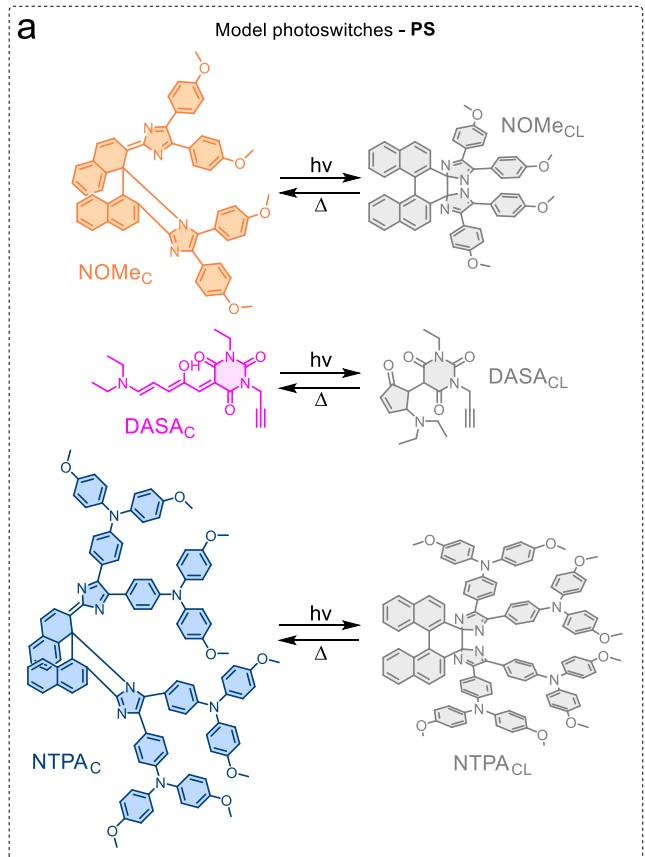

**Fig. 2 | Structures of the model monomers employed in this work. a** Structure of the model photoswitches PS and their respective isomerization. The colored FRET-active forms of **NOMe, DASA,** and **NTPA** are represented in orange, pink, and blue, respectively. The colorless forms are represented in gray. **b** Structures of the model fluorophores 2PAP.

## Results and discussion

### Design approach

A conventional two-photon absorbing fluorescent probe (2PAP) emits with an intensity that is proportional to the square of the excitation intensity: $I_{em} \propto I_{exc}^2$. Hence, if we successfully intertwine two excited-state processes, both triggered by 2PA, where one is producing fluorescence, it should be possible to observe a quartic dependence of the type $I_{em} \propto I_{exc}^4$ without the need of using a 4PA fluorophore.

Our design idea builds on the electronically orthogonal coupling of a PS showing T-type negative photochromism into a molecular dyad or triad with a 2PAP (Fig. 1). This ensures that the colored form of the photoswitch ($PS_C$) is the thermally stable form and that the chromophoric moieties preserve their respective spectroscopic identities.

By carefully selecting both the PS and the 2PAP components (Fig. 2), the dyad can be engineered such that, in its $PS_C$–2PAP form, the fluorescence of the 2PAP is quantitatively quenched via Förster resonance energy transfer (FRET) to the colored form of the photoswitch $PS_C$. In consequence, the FRET populates the excited state of the PS, which subsequently isomerizes to the colorless form ($PS_{CL}$). In effect, two-photon excitation of the 2PAP sensitizes the

**Table 1 | Properties of the model compounds in air-equilibrated toluene solutions, unless otherwise noted**

| | $\lambda_{abs}$ (nm)[a] | $\varepsilon/10^3$ (M$^{-1}$ cm$^{-1}$)[b] | $t_{1/2}$ (s)[c] | $\Phi_{iso}$ [d] | $\lambda_{em}$ (nm)[e] | $\Phi_F$ [f] | $\tau_F$ (ns)[g] | $\sigma_{2PA}$ (GM)[h] |
|---|---|---|---|---|---|---|---|---|
| **NOMe** | 490[i] | 35.0[j] | 110 | 0.03 | | | | |
| **DASA** | 570[k] | 123 | 24 | 0.11 | | | | |
| **NTPA** | 577[l] | 35.0[m] | 15 | 0.005 | | | | |
| **Napht** | 402 | 11.2 | | | 489 | 0.80[n] | 7.4 | 14 |
| **FRT** | 403 | 60.0[o] | | | 465 | 0.93 | 2.1 | 250[o] |
| **Phtha** | 385 | 6.0[p] | | | 461 | 0.59[q] | 13.4 | 7 |

[a] Longest-wavelength absorption maximum; in the case of a PS, the colored form is considered.

[b] Molar absorption coefficient; in the case of a PS, the colored form is considered.

[c] Thermal half-life of the colorless form in benzene (**NOMe** and **NTPA**) or toluene (**DASA**).

[d] Photoisomerization quantum yield in toluene; in the case of **DASA**, corresponding to the actinic step of a multistep ring-closing mechanism.

[e] Emission maxima upon excitation at 365 nm.

[f] Fluorescence quantum yield.

[g] Fluorescence lifetime.

[h] 2PA cross section at 800 nm in toluene (**Napht, Phtha**) or ethanol (**FRT**).

[i] Consistent with literature data for the same compound in benzene solution; ref. 33.

[j] In benzene, taken from ref. 36.

[k] Consistent with literature data for a related *N*-methylated derivative in toluene solution reported in refs. 30 and 49.

[l] The longest-wavelength absorption maximum for this compound in benzene solution was reported as 575 nm; see ref. 34.

[m] In benzene; taken from ref. 34.

[n] In benzene; see ref. 36.

[o] In ethanol solution; see ref. 35.

[p] Reported for a similar derivative in DMSO solution in ref. 37.

[q] In benzene; see ref. 38.

photoswitching[22,27,28]. Direct excitation of PS$_C$, if any, will not compromise the 4for2 function as it would trigger the same isomerization process to yield the desired PS$_{CL}$ form (see Supplementary Note 1)[29]. In the resulting PS$_{CL}$−2PAP state, the FRET-induced emission quenching of the 2PAP would be deactivated due to the zero spectral overlap between the 2PAP fluorescence and the PS$_{CL}$ absorption. Thus, the next two photons that arrive at the fluorophore would trigger its fluorescence. In essence, we would have a stepwise mechanism in which both the fluorescence intensity "per fluorescent molecule" and the concentration of the fluorescent molecules depend quadratically on the excitation intensity. This results in the desired overall quartic dependence of the fluorescence intensity with the excitation intensity, albeit only 2PA processes participated.

It is important to note that once FRET has fully converted the PS from its colored form PS$_C$ to the colorless PS$_{CL}$ state, the system becomes saturated, and the fluorescence reverts to the standard quadratic dependence on excitation intensity, corresponding to a single 2PA process[22]. This occurs as there is no longer any dependence of the PS$_{CL}$−2PAP concentration on the excitation intensity. An illustrative example of this saturation effect is a previous work of ours where a P-type diarylethene was used as the photoswitch[22]. However, if the PS$_{CL}$ form can thermally revert to the PS$_C$ form, a dynamic photothermal equilibrium is established where a low steady-state concentration of fluorescent PS$_{CL}$−2PAP is maintained. This is ensured by using a T-type negative PS, which recycles the thermodynamically more stable PS$_C$−2PAP form in a thermally activated process. The effective steady-state concentration of the two isomeric forms depends on the delicate balance between the (macroscopic) rates of the photoisomerization ($k_1$) and the thermal back isomerization ($k_{-1}$), where $k_{-1}$ should outnumber $k_1$ to guarantee a high [PS$_C$−2PAP]/[PS$_{CL}$−2PAP] ratio (Fig. 1). Such a system would constitute a paradigm shift for multiphoton-excitable fluorophores with wide-reaching consequences for their application potential in imaging and microscopy.

## Implementation

To implement our design concept at the molecular level, we required a T-type negative PS and a spectrally compatible 2PAP. The 2PAP fluorescence should show exclusive spectral overlap with the absorption spectrum of the colored, thermodynamically stable form PS$_C$. After a literature search, we settled for the PS and 2PAP chromophores shown in Fig. 2. The PS include a 1$^{st}$ generation donor-acceptor Stenhouse adduct (**DASA**)[26,30–32] and 1,1′-binaphthyl-bridged imidazole dimers with varying substitution patterns (**NOMe** and **NTPA**)[33,34]. The 2PAP moieties are push-pull architectures based on a donor-acceptor substituted fluorene (**FRT**)[35] or *N, N*-dimethylamino-substituted aromatic dicarboximides (**Napht** and **Phtha**)[36–38]. The photophysical properties of the individual 2PAP and the PS are compiled in Table 1. The chromophoric building blocks were synthetically linked together, yielding the dyad and triad structures shown in Fig. 3 (see the Supplementary Information for details about the synthesis and compound characterization).

1,1′-Binaphthyl-bridged imidazole dimers, originally developed by Abe[25], are T-type negative PS that can be reversibly toggled between a thermally stable colored form and a colorless isomer. Their absorption spectra can be tuned by means of the modification of electron-donor substituents[33,34,39], and they offer multiple modification sites for the integration of the 2PAP. These features have been exploited to prepare the dyad ***asy*NOMe/Napht** as well as the triads **NOMe/Napht, NTPA/Napht**, and **NOMe/Phtha** (Fig. 3). The PS exhibit high molar absorption coefficients ($\varepsilon$ *ca.* 35000 M$^{-1}$ cm$^{-1}$), while the unquenched 2PAP components show high fluorescence quantum yields ($\Phi_F$ 0.59 and 0.80 for the employed aromatic dicarboximides **Phtha** and **Napht** in toluene, respectively). These properties predict very efficient FRET with the colored forms of the PS for all tested combinations (see below and Table 2; more details are available in the Supplementary Information). In addition, the thermal back-isomerization from the colorless to the colored form of the PS occurs on the timescale of a few seconds (Table 1), rendering these PS ideally suited for the purpose of 4for2.

To illustrate the experimental realization of the above design principles, we focus on the **NTPA/Napht** triad as a representative example. Fig. 4a shows the absorption spectra of **NTPA** and **Napht** alongside the fluorescence emission of **Napht**. The data clearly shows a substantial spectral overlap between the emission of the 2PAP and the absorption of the colored form of the PS (**NTPA$_C$**). In contrast, the colorless photoisomerized **NTPA$_{CL}$** absorbs only marginally at wavelengths longer than 450 nm, resulting in negligible spectral overlap with the **Napht** emission. Fig. 4b displays the absorption and emission

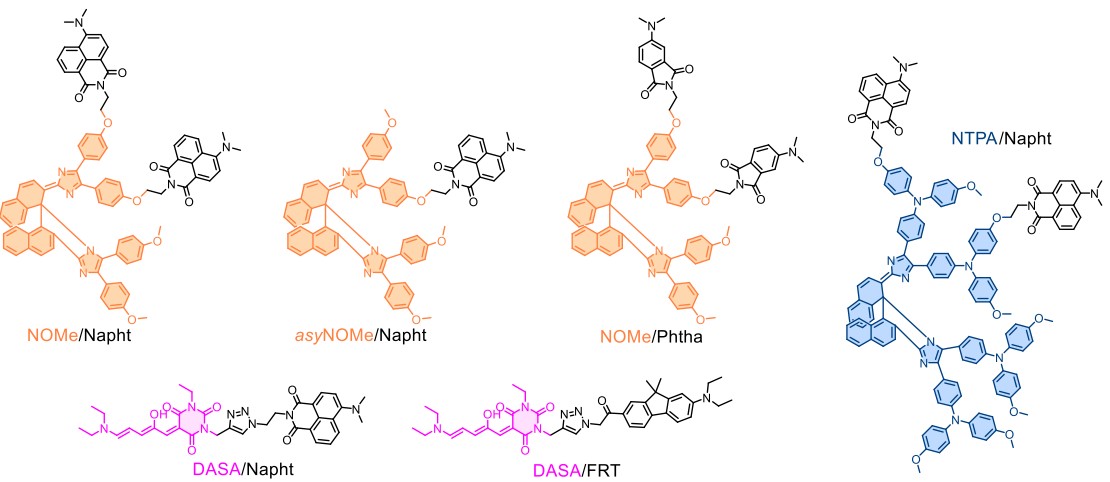

**Fig. 3 | Structures of the dyads/triads synthesized and characterized in this work.** The photoswitches are represented in their respective FRET-active forms and colored accordingly (**NOMe**: orange, **DASA**: pink, **NTPA**: blue).

**Table 2 | Photophysical properties of the dyads and triads in air-equilibrated toluene solutions**

|  | $\lambda_{abs}$ (nm)[a] | $\lambda_{em}$ (nm)[b] | $\tau_F$ (ps)[c] | $E_{FRET,theor}$[d] | $E_{FRET,exp}$[e] | Slope log-log plot[f] | FR[g] |
|---|---|---|---|---|---|---|---|
| **NOMe/Napht** | 489 | 493 | 5.5 | 0.999 | 0.999 | 3.80 ± 0.45 | 97% |
| **asyNOMe/Napht** | 489 | 493 | 3.1 | 0.999 | 0.999 | 3.17 ± 0.42 | 98% |
| **DASA/Napht** | 571 | 499 | 4.1 | 0.999 | 0.999 | 3.78 ± 0.24 | 99% |
| **NTPA/Napht** | 577 | 492 | 3.9 | 0.999 | 0.999 | 3.94 ± 0.27 | 96% |
| **DASA/FRT** | 571 | 467 | 1.7 | 0.999 | 0.999 | 3.84 ± 0.16 | 97% |
| **NOMe/Phtha** | 490 | 462 | 1.4 | 0.999 | 0.999 | 4.46 ± 0.39 | 97% |

[a] UV/vis absorption maximum of the colored form ($PS_C$).
[b] Emission maximum of the 2PAP fluorophore.
[c] Fluorescence lifetime of the quenched emission in $PS_C$–2PAP. Monoexponential decays, except for **NOMe/Phtha**, where an amplitude-averaged lifetime is given [$\tau_1$ = 0.43 ps (71%) and $\tau_2$ = 3.8 ps (29%)].
[d] Theoretically expected FRET efficiency, based on the critical FRET radius $R_0$ and actual interchromophore distance $R$; see the Supplementary Information for details.
[e] Experimentally observed FRET efficiency; $E_{FRET}$ = 1 − $\tau_F$($PS_C$-2PAP)/$\tau_F$(2PAP model).
[f] Slope of the log-log plot of fluorescence intensity versus excitation intensity.
[g] Fatigue resistance FR; shown is the percentage of the intact population after ten switching cycles.

spectra of the **NTPA$_C$/Napht** triad. The comparison of both sets of spectra confirms that the chromophores behave as electronically decoupled moieties in the triad (also confirmed by TDDFT calculations; see the Supplementary Information). This finding demonstrates that the individual spectral properties of the building blocks are preserved in the conjugate, as further detailed in Tables 1, 2.

When comparing the fluorescence lifetimes of the model chromophore **Napht** and the thermally stable **NTPA$_C$/Napht** isomer of the triad, a dramatic quenching is noted. The fluorescence decay of the triad features one component of 3.9 ps, while the **Napht** model alone shows a mono-exponential decay with a lifetime of 7.4 ns. These results, obtained from femtosecond fluorescence up-conversion and time-correlated single-photon counting measurements, are presented in detail in the Supplementary Information. This corresponds to nearly quantitative fluorescence quenching (> 99.9%). As inferred from the critical FRET radius $R_0$ (48 ± 2 Å), such a quantitative FRET process is expected (Table 2) for the triad with an actual center-to-center interchromophore distance of 17.5 Å (estimated from the energy-optimized calculated structure); see the Supplementary Information. Noteworthy, the FRET-induced quenching applies for conventional one-photon excitation as well as for two-photon absorption, both resulting in an identical excited singlet state of the 2PAP. Furthermore, the push-pull aminonaphthalimide **Napht** exhibits an appropriate 2PA cross section of approximately 14 GM at an excitation wavelength of 800 nm

(see 2PA spectra in the Supplementary Information). While the fluorescence of **Napht** is quenched due to FRET to the colored **NTPA$_C$** form, it is visible to the full extent in the **NTPA$_{CL}$/Napht** triad due to the absence of energy transfer.

Photoswitching between the colored **NTPA$_C$** and the colorless **NTPA$_{CL}$** forms is initiated by direct excitation with visible light that targets the π,π*-absorption band, typically with a quantum yield of about 0.005[34]. An alternative pathway involves FRET-sensitization via two-photon excited 2PAP, which is key to our approach. Once the photoswitch is converted to its colorless form, it spontaneously reverts back to the colored form through a thermally activated dark process with a half-life of roughly 15 s for **NTPA$_{CL}$**[34]. This reversible behavior is crucial for achieving the 4for2 function, as it maintains a low steady-state concentration of the fluorescent **NTPA$_{CL}$/Napht** species and prevents saturation effects. Moreover, the triad exhibits robust, fatigue-resistant photoswitching, with 96% of the initial molecules remaining intact after ten cycles of alternating visible-light irradiation and thermal back-isomerization (Table 2 and the Supplementary Information for details).

All performance aspects that were discussed above for the **NTPA/Napht** triad can be seamlessly transferred to the other herein investigated dyads and triads (see experimental data in Table 2 and in the Supplementary Information). This consistency highlights the molecular flexibility of our approach within the photophysical

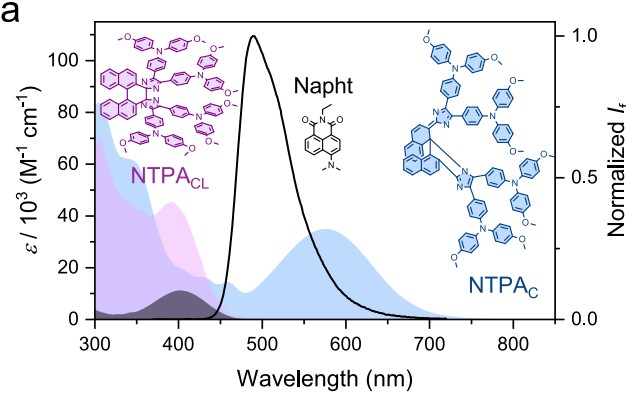

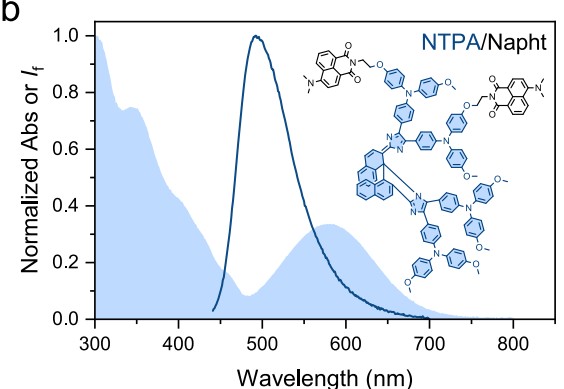

**Fig. 4 | Absorption and emission spectra of model compounds Napht and NTPA vs. triad NTPA/Napht. a** Absorption (filled areas) and emission (solid line) spectra in toluene of model compounds **NTPA_C** (blue), **NTPA_CL** (purple), and **Napht** (black). Due to the efficient thermal isomerization to yield **NTPA_C**, we could not completely convert the sample to the **NTPA_CL** isomer. The absorption spectrum of **NTPA_CL** was therefore obtained by deconvolution analysis. **b** Absorption spectrum of **NTPA_C/Napht** triad (filled area) and emission spectrum (solid line) registered at a certain photothermal distribution upon irradiating at 405 nm a solution containing the colored species **NTPA_CL/Napht**.

boundaries that define the fundament of 4for2 (see below). These include not only combinations of 2PAP with 1,1′-binaphthyl-bridged imidazole dimers, but also other T-type negative PS such as **DASA** (e.g., in **DASA/FRT**).

**Validation**

Having established the two entangled 2PA-induced processes (the 2PA-induced FRET, leading to sensitized **NTPA_C → NTPA_CL** photoisomerization and the 2PA-induced fluorescence of the 2PAP moiety), as well as the occurrence of competitive thermal back-isomerization, we aimed for demonstrating the desired 4for2 feature. Fig. 5a displays the fluorescence intensity of **NTPA/Napht** at equilibrium for irradiation with varying excitation pulse intensities of 800-nm light. The log-log plot of the emission intensity versus the excitation pulse energy (Fig. 5b) yields a straight line with a slope of $3.94 \pm 0.37$ (Table 2), being in excellent agreement with the expected quartic dependence (slope of 4) characteristic of a four-photon-induced fluorescence. All other herein investigated combinations of PS and 2PAP feature a similar behavior with slopes varying between 3.17 and 4.46 (Table 2 and the Supplementary Information; see also Supplementary Note 2). The combined observations showcase the validity of the 4for2 approach, enabling the use of 2PA to generate potentiated nonlinear fluorescence emission as would correspond to a 4PA fluorophore.

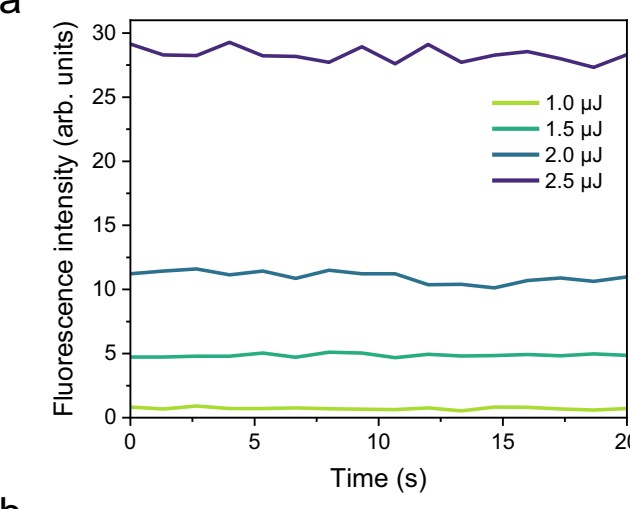

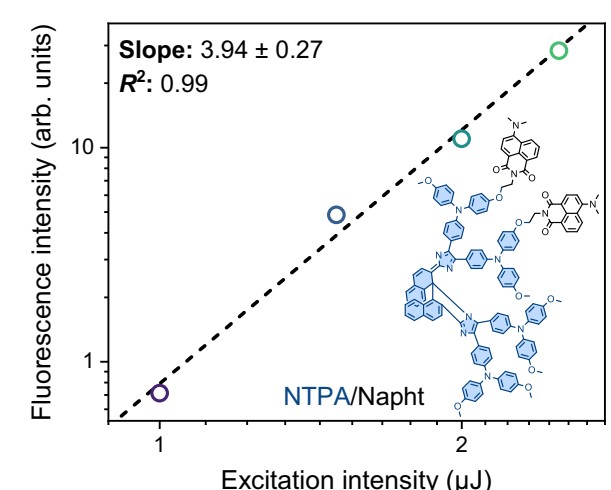

**Fig. 5 | Validation experiments for NTPA/Napht under two-photon excitation. a** Fluorescence intensities of **NTPA/Napht** in air-equilibrated toluene, monitored for 20 s at equilibrium upon excitation at 800 nm at varying excitation intensities. **b** The corresponding log-log plots.

To conclude, we have shown that the concatenation of two 2PA processes in photoswitch-fluorophore constructs allows for the observation of a quartic dependence of the emission intensity on the excitation light intensity. To implement the required photophysical boundary conditions, T-type negative PS were combined with 2PAP. In this system, two-photon-induced photoswitching, sensitized by FRET from the 2PAP to the colored form of the PS, provided the colorless isomer. The absence of FRET in the latter state made it possible to observe the two-photon-excited fluorescence of the 2PAP. Hence, the 2PAP is involved in two consecutive 2PA processes. Indeed, this entanglement of two 2PA processes resulted in a nearly ideal behavior in accordance with the predicted quartic dependence.

The use of T-type switching is instrumental as it provides an inherent process to guarantee the spontaneous reversibility needed for observing a stable fluorescence response over time. In principle, one could also imagine P-type PS for the same purpose, which, however, would have to draw on technologically much more cumbersome two-color laser experiments. The proof-of-concept of the herein developed 4for2 approach provides molecular systems with nonlinearly potentiated fluorescence without having to resort to 4PA fluorophores. This is significant, as 4PA fluorophores are much less common compared to the vast array of 2PA fluorophores available.

The transition to biologically more relevant environments will require the careful design of molecular systems that meet the application-specific criteria (water solubility, polarity dependent photochemical/photophysical properties etc.). Nonetheless, the 4for2 approach showcases a paradigm shift towards the wider implementation of advanced multiphoton microscopy imaging, building on molecular tools (T-type negative PS and 2PAP) that are readily available to the community.

## Methods

### General methods and materials for synthesis

Commercially available reagents and solvents were purchased and used as supplied, unless otherwise stated. Dry tetrahydrofuran (THF), toluene, dichloromethane (DCM), acetonitrile (ACN) and dimethylformamide (DMF) were obtained from a solvent purification system (PS-MD-5/7 Inert technology).

Microwave reactions were performed in a Biotage Initiator Reactor using single-mode irradiation, with controlled temperature and pressure, and with fixed hold time.

Reactions were monitored by analytical thin-layer chromatography (TLC) on silica gel plates (Merck Kieselgel 60, $F_{254}$) and revealed by using a UV lamp (excitation at 254 or 365 nm).

Flash-column chromatography was performed either on a Biotage Isolera or Biotage Selekt flash chromatography systems, using pre-fabricated silica columns: Sfär Silica D Duo 60 μm, Sfär Silica HC High Capacity 20 μm or Sfär KP-Amino D Duo 50 μm cartridges for normal-phase, and Sfär C18 D Duo 100 Å 30 μm cartridges for reverse-phase.

NMR spectra were recorded at room temperature using various spectrometers: an Agilent 400 MHz NMR spectrometer operating at 400 and 101 MHz for $^1$H and $^{13}$C, respectively; a Bruker Avance NEO spectrometer, operating at 600 and 151 MHz for $^1$H and $^{13}$C, respectively; a Bruker Avance III HD spectrometer with an Oxford 800 magnet, operating at 800 and 201 MHz for $^1$H and $^{13}$C, respectively; a Bruker Avance III HD spectrometer with an Oxford 900 pumped magnet, operating at 800 and 226 MHz for $^1$H and $^{13}$C, respectively. Chemical shifts (δ) are reported in parts per million (ppm) with reference to the residual protic solvent of $CDCl_3$, $CD_2Cl_2$, DMSO-$d_6$ or $C_6D_6$. Abbreviations for the multiplicity in $^1$H NMR spectra are as follows: s = singlet, d = doublet, t = triplet, q = quartet, m = multiplet. Coupling constants ($J$) are reported in Hz.

HRMS data were recorded with a QExactive HF Orbitrap mass spectrometer interfaced with a Dionex Ultimate 3000 liquid chromatography system (Thermo Fisher Scientific). The instrument operated in full MS mode only, where the ion mass spectra were acquired at a resolution of 120,000, maximum injection time 200 ms for 3e6 ions. The Orbitrap was calibrated with Pierce LTQ ESI Positive Ion Calibration Solution prior to the analysis, resulting in mass accuracy better than 5 ppm. Electrospray ionization was performed at 4 kV and 320 °C using a metal emitter in the ion source. The sample (1 or 10 μL) was injected onto a reversed-phase XBridge BEH C18 column (3.5 μm, 2.1 × 50 mm, Waters). The analysis was performed using a linear gradient over 2.5 min from 10 to 100% solvent B, followed by isocratic elution with 100% solvent B for 17.5 min with a flow of 0.300 mL/min (*solvent A*: $H_2O$ with 0.1% formic acid; *solvent B*: 80% ACN in $H_2O$ with 0.1% formic acid). Data analysis was performed using the Xcalibur software (Thermo Fischer Scientific).

### General methods and materials for optical spectroscopy

Spectroscopic grade solvents were employed for all photophysical studies.

Optical spectroscopy experiments were performed at room temperature, with air-equilibrated optically diluted solutions (absorbance ≤ 0.1, implying concentration equal to or below 10 μM) or 10 μM solutions, using 10 mm × 10 mm pathlength quartz cuvettes, unless otherwise noted.

Ground state absorption spectra (UV/vis) were recorded on a Cary 50 UV-vis-NIR spectrometer. Corrected fluorescence spectra were recorded on a SPEX Fluorolog-3 spectrofluorometer.

Fluorescence lifetimes at the nanosecond time scale were determined using the time-correlated single-photon counting (TC-SPC) technique. The frequency-doubled output (360-400 nm, 81 MHz, 2 ps FWHM) of a mode-locked Ti:Sapphire laser (Tsunami, Spectra Physics) was used as the excitation source. The repetition rate of the laser system was reduced to 8.1 MHz by a pulse picker (Spectra-Physics). The linearly polarized excitation light was rotated to a vertical plane using a Berek compensator (New Focus) in combination with a polarization filter and directed onto the sample. The emission was collected at a 90° angle relative to the incident light and guided through a polarization filter set to the magic angle (54.7°) with respect to the polarization plane of the excitation beam. The fluorescence was spectrally resolved using a double monochromator (Sciencetech 9030, 100 nm focal length, wavelength accuracy 0.3 nm) and detected by a microchannel plate photomultiplier tube (MCP-PMT, R3809U-51, Hamamatsu). A time-correlated single-photon timing PC module (SPC 830, Becker & Hickl) was employed to obtain the fluorescence decay histogram in 4096 channels. The decays were recorded with 10000 counts in the peak channel, using time windows of 30–60 ns, and analyzed individually with time-resolved fluorescence analysis (TRFA) software based on iterative reconvolution of the data with the instrumental response function (IRF). The full width at half maximum (FWHM) of the IRF was typically on the order of 42 ps.

The characterization of the two-photon absorption properties of **Napht** and **Phtha** was carried out on a Zeiss LSM 710 epi-fluorescence microscope equipped with an InSight DeepSee (Spectra-Physics) tunable wavelength laser system, delivering 100-ps pulses at 80 MHz. The compounds were dissolved in toluene, diluted to a concentration of 10 μM and then placed inside 5 μL microcapillary tubes (Hirschmann) and sealed. Excitation and emission light were transmitted through a Plan-Apochromat 10x/0.45, ∞/0.17 objective with a working distance of 2.0 mm and focused at the air/liquid boundary, allowing the simultaneous detection of sample and background fluorescence. Emission spectra were measured in a laser power regime where the fluorescence was proportional to the square of the laser excitation power and using a dynamic wide emission detection window between 420–670 nm in intervals of 10 nm or 3 nm. Emission spectral data for compound and background regions of interest (ROIs) were registered using ImageJ software[40,41]. The two-photon absorption cross sections were determined by the two-photon induced fluorescence method[5,42] and according to Eq. (1):

$$\sigma_{2PA,s} = \sigma_{2PA,r} \frac{C_r n_r \Phi_{f,r} F_s}{C_s n_s \Phi_{f,s} F_r} \qquad (1)$$

where $\sigma_{2PA}$ is the two-photon absorption cross-section, $C$ is the concentration of the species, $\Phi_f$ is the fluorescence quantum yield, $n$ is the refractive index of the used solvent, and $F$ is the integrated emission spectrum recorded at each wavelength. The letters $s$ and $r$ are used to denote *sample* and *reference*, respectively. Rhodamine 6 G (**Rho6G**) in methanol ($\Phi_f$ = 94%, $\sigma_{2PA,r}$ data adopted from ref. [43]) was used as reference, assuming that fluorescence quantum yields remain the same regardless of the excitation mechanism. To determine the multiplicity of the multiphoton absorption process, emission was recorded at different excitation intensities, and the integrated emission in the emission wavelength range was plotted as a function of relative excitation intensity.

Fatigue resistance was assessed by monitoring changes in the UV/vis absorption of the dyads and triads in air-equilibrated toluene solution over ten isomerization cycles, each of these consisting of irradiation with a 405 nm LED as light source until reaching *ca.* 10-20%

conversion to the colorless species, followed by thermal back isomerization to the initial form.

**Experimental details for femtosecond fluorescence up-conversion.** The samples were prepared in solution and contained in a 1 mm quartz cuvette (absorbance around 1 at the excitation wavelength). During the measurements, the samples were kept fixed but repositioned before each recording of the traces to ensure a fresh spot was excited at the beginning of each experiment. The signal was not averaged.

An amplified femtosecond double OPA (optical parametric amplifier) 35 fs-laser system was used to provide excitation pulses of 360−400 nm. The power of the excitation beam was set to 300 μW, and the fluorescence light emitted from the samples was efficiently collected using a parabolic mirror. The fluorescence was then filtered using a 420 nm long pass filter to suppress the scattered light, directed, focused and overlapped with a gate pulse (800 nm, *ca.* 10 μJ) derived from the regenerative amplifier onto an LBO crystal. By tuning the incident angle of these two beams relative to the crystal plane, the sum frequency of the fluorescence light and the gate pulse was generated. The time-resolved traces were then recorded by detecting the sum frequency light while changing the relative delay of the gate pulse versus the sample excitation time. Fluorescence gating was done under magic angle conditions in time windows of 6 and 50 ps.

Monochromatic detection was carried out in heterodyne mode using a PMT (R928, Hamamatsu) placed at the second exit of the spectrograph mounted behind a slit. Optical heterodyne detection is a highly sensitive technique that is used to measure very weak changes in absorption induced by a frequency-modulated pump beam. An additional bandpass filter 260−380 nm was placed in front of the monochromator to reject excitation light and the second harmonic of the gate pulse. The electrical signal from the photomultiplier tube was gated by a boxcar averager (SR 250, Stanford Research Systems) and detected by a lock-in amplifier (SR830, Stanford Research Systems). The prompt response (or instrumental response function, IRF) of this setup (including laser sources) was determined by detection of scattered light of the excitation pulse under identical conditions and found to be approximately 100 fs (FWHM). This value was used in the analysis of all measurements for curve fitting using iterative reconvolution of the data sets while assuming a Gaussian shape for the prompt response.

Global analysis of the fluorescence decays obtained at different wavelengths as a sum of exponentials allowed us to obtain amplitude-to-wavelength spectra. When interpreting the features of such spectra, one should keep in mind that no correction for the wavelength dependency sensitivity of the detection setup (mixing crystal, filters, PMT) was applied. The samples were prepared in solution and contained in a quartz cuvette. To improve the signal to noise ratio, every measurement was averaged 15 times at 128 delay positions where a delay position is referred to as the time interval between the arrival of the pump and gate pulses at the sample position.

**Experimental details for excitation intensity dependence fluorescence / two-photon validation studies.** The samples were prepared and treated as described above for the femtosecond fluorescence up-conversion experiments.

The same amplified femtosecond double OPA laser system was used to provide excitation pulses at 800 nm. The pulses were directly derived from the regenerative amplifier (35 fs pulse duration). However, their peak power and duration were modified to 3 ps by passing the beam through a 100 mm thick BK7 glass. The energy per pulse of the excitation beam was gradually adjusted using a polarizer (New Focus), then focused onto the sample using a 400 mm focal length concave mirror. The beam waist diameter at the focal plane was approximately 100 μm. The fluorescence light emitted by the samples was efficiently collected from the opposite side of the excitation beam using a 300 mm focal length lens. The fluorescence was then filtered using a 600 nm short-pass filter to suppress scattered light.

Monochromatic detection in heterodyne mode was performed using a PMT (R928, Hamamatsu) placed at the second exit of the spectrograph, mounted behind a slit. A modified optical heterodyne detection method was employed to measure very weak emission induced by the excitation pulses. The electrical signal from the photomultiplier tube was gated by a boxcar averager (SR 250, Stanford Research Systems) and detected by a lock-in amplifier (SR830, Stanford Research Systems). The prompt response (or instrumental response function, IRF) of this setup, including laser sources, was estimated to be about 1 s, determined primarily by the time constant of the lock-in amplifier and the data reading rate.

### General methods for theoretical calculations

Gaussian 16C01[44] was employed for the theoretical calculations.

The geometrical parameters for the ground state ($S_0$) were determined with the density functional level of theory (DFT), employing the *m*PW1PW91 functional[45] and the 6−31+G(d) basis set. Solvent effects were considered by including the solvation model based on density (SMD)[46]. The absolute nature of the energetic minima was established by the absence of a negative frequency in the vibrational analysis.

Energy parameters were calculated as vertical electronic excitations from the $S_0$ minima structure using the linear response (LR) approach and the time-dependent density functional response theory (TDDFT)[47]. These calculations were carried out for the first fifteen excited states (twenty in case of **NTPA/Napht**) at the SMD(toluene)/*m*PW1PW91/6-31+G(d) level.

Natural Transition Orbitals (NTOs) were analyzed to confirm the absence of electronic interactions between the building blocks integrated onto the dyads and triads. These orbitals result from combining proportionally the different elementary orbitals that participate in an electronic transition, thus giving a better description of their nature[48].

## Data availability

The datasets generated during and/or analyzed during the current study are available in the KU Leuven RDR repository, https://doi.org/10.48804/WHKQSH. All data is available from the corresponding author upon request.

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

## Acknowledgments

The authors are indebted to the European Innovation Council and SMEs Executive Agency (EISMEA) for financial support of the EIC PATHFINDER (project 101098934 – 4for2). J. A. also thanks the Swedish Research Council VR (project 2021-05311). In memory of Prof. Miguel A. Miranda.

## Author contributions

C.B.-M. and J.R. designed, synthesized, and characterized all compounds and conducted their fundamental spectroscopic characterization. C.B.-M. performed DFT and TDDFT calculations and investigated their two-photon absorption properties. E.F., F.d.J., J.A., and C.B.-M. carried out the ultrafast spectroscopy studies and the validation experiments under two-photon excitation. M.G., J.H., U.P., and J.A. conceived and supervised the work. The paper was written by U.P., J.A., and C.B.-M. with input from all authors. All authors contributed to discussions.

## Funding

## Competing interests

J. A. is the inventor on a granted patent (Patent No. US11698343B2) held by J. A., which covers the generic design and intended function of the compounds described in the manuscript. J. A. declares no other competing interests. C. B.-M., J. R., E. F., F. d. J., M. G., J. H., and U. P. declare no competing interests.
