## [Transparent Peer Review file · Nature Communications]

Exploiting negative photochromism to harness a four-photon-like fluorescence response with two-photon excitation

Corresponding Author: Professor Joakim Andreasson

Version 0:

Reviewer comments:

Reviewer #1

(Remarks to the Author)

The main contribution of this manuscript is the realization of $n > 2$ photon dependencies by sequentially actuating in a stepwise manner (two photons at a time) on the molecular activation and photoswitching properties. This is based on a modular scheme where a two-photon-active fluorophore is deactivated by FRET by a secondary chromophore which can be transformed into its negatively photochromic state (which is not energy accepting). Once this secondary chromophore is switched, the energy transfer channel is deactivated, thereby "allowing" the two-photon chromophore to have a high emission yield. A critical issue is a low-concentration of the fluorescent state which in turn depends on the rate of re-conversion from the photoswitched state to the initial state. This is an interesting innovative scheme. I recommend publication after minimal issues are considered by the authors.

Minor issues

Should comment on the two photon cross sections of the PS themselves, in some cases, the PAP systems show relatively low tpa cross sections (of less than 20 GM). It is possible that from the conjugation of the PS, they themselves, could have competing two-photon absorption (this could be a positive characteristic as the objective of the switching may not require that the two-photons are absorbed by the PAP systems, but by the PS themselves).

The authors should include experiments where (possibly at low temperatures where the thermal back isomerization is inhibited) it is shown which is the increase in the steady state emission from the 2PAS as the system evolves into a photo-stationary state. This would directly indicate which is the amount of fluorescence present in the system before the photo-transformation takes place, showing the dynamic range for the emission-turn-on. Although this is presented in terms of emission lifetimes, the steady state emission increase at the focal point of the two-photon system would be interesting.

The authors should consider giving more molecular information right in the abstract of the molecules which are the actuators in these designs. The general concepts are well described, but to several areas, more rapid info about which are the actual molecular entities and how their intrinsic properties add to their combined value would be of great value in the abstract itself.

It is not uncommon that non-linear excitation schemes show dependencies that are larger than square due to 3PA, etc. Would it be possible to show that in same concentrations with the same 2PAPS, the power dependence turns from square to quartic (for the dyads) in comparison?

The general concepts related to a sequential set of effects (each with 2P dependence) is clever and of significant potential value. I think the authors should also be explicit about the challenges that this kind of implementation will face. Some comments about this will help future researchers about additive contributions in this line. For example, which are the challenges in terms of solubility and how the photochromic and thermal dynamics will be affected by the solvent (for example, in DASAs this is highly affected by the solvent). Different negative-photoswitches will have quite different behaviors in different biological environments.

Reviewer #2

(Remarks to the Author)

This manuscript presents a very interesting study about the development of chemically linked dyads and tryads composed of 2PA fluorophores and T-type photoswitches. The experiments were well performed and the results are convincing. A lot of data is presented and discussed with an overall demonstration of quartic excitation dependence for 6 different dyads and tryads composed of 3 different 2PA fluorophores and 3 different T-type photoswitches (why were those 6 selected out of the 9 (or more when considering the dyads and tryads) possible combinations?). This is very impressive work that significantly extends beyond the authors' previous work using a dyad with a 2PA fluorophore and a P-type photoswitch. The proof-of-principle demonstration has the potential to open many powerful experimental possibilities for imaging with high spatial resolution and therefore, this manuscript is well suited for Nature Communications. I recommend the publication of this manuscript after revision (see comments below) to make the results easier to understand and to clarify some parts of the study that may not be clear to everyone (including me...).

Comments:

I would recommend to clearly mention the differences compared to the authors' previous work with the P-type PS dyads. Considering the broad readership of Nat Comm, the differences (which are significant) may not be obvious to everyone.

Figure 3 should also show the absorption of NTPA(CL), such that the missing (or negligible) spectral overlap can be appreciated. It could be added as a graph c (with same wavelength range) below the other two to avoid too many spectra in one graph.

P11: The authors suggest that the spectra show an electronically decoupled system. However, it seems that the blue part of the composed spectrum is higher (comparing for example the intensities at 300 and 600 nm for both absorption spectra) than would be expected from a 2:1 linear combination in the tryad system. Are there maybe free Napht in the system or some parts of NTPA in the tryad are already in the colorless form (which would show stronger blue absorption – another reason to show that spectrum in Figure 3)?

P12: How was the Förster distance of 4.82 nm calculated? I also do not think that a 3-digit accuracy is realistic. Please change to 4.8 nm and provide an error estimation. Based on overlaps and uncertainties with spectra in different conditions and solvents, I would expect 4.8 +/- 0.5 nm or alike. But maybe I am wrong...

It seems that the FRET analysis was only performed with time-resolved spectroscopy (comparing lifetimes of D and DA). What about the fluorescence spectra of D and DA? They are mentioned but not shown (or maybe they are somewhere in the 100 pages SI). I assume that one cannot see much (if the quenching is really close to 100%) but it would be nice to have the spectra next to the lifetime data (in the SI section 4) and also analyze the intensities for FRET efficiency determination.

Validation: Wait a minute... Would two entangled 2PA processes not lead to a slope of 16 when there is quartic dependence (2^4)? The slope of 4 shows a quadratic dependence (2^2), which would mean the authors showed 2 for 2 and not 4 for 2? Or is that the main difference compared to their previous dyad? Would that not also mean that the previous dyad (which had a slope of 10) would be much better when it comes to spatial resolution?

Conclusion: "In principle, one could also imagine P-type PS for the same purpose...". But that was already shown in their previous dyad, right? And no two-color excitation was necessary.

Reviewer #3

(Remarks to the Author)

This study reports a clever design of fluorescent dyes for "four-photon-like" absorption. For this purpose, the authors coupled fluorophores to photoswitches that show a rather fast thermal relaxation upon photoisomerization. The stable colored isomer of the photoswitch quenches the fluorescence emission, while the metastable non-colored isomer does not. In this way, a two photon excitation leads to a quartic dependence of the emission intensity on the excitation intensity.

Since the paper contains experiments on a set of dyes combining different photoswitches and fluorophores, it is likely that the concept can be generalized to other dyes with matching spectra and fast relaxation of the photoswitch.

This paper is highly relevant since the concept described here can be used to enhance the spatial resolution of optical microscopy without unwanted side effects such as irreversible saturation.

The experimental work is rigorous and well-described. Obviously the impact of the paper would be significantly increased if the authors included a proof of concept of a microscopy experiment, however in my view the current manuscript with its focus on molecular design and spectroscopy merits publication.

Version 1:

Reviewer comments:

Reviewer #1

(Remarks to the Author)

I consider that the issues regarding the reviewers comments have been addressed, I recomend publication.

Reviewer #2

(Remarks to the Author)

The authors have very carefully revised their manuscript and looked into many details of their results to reply to the reviewers' concerns. The authors have also improved their manuscript by considering all reviewers' comments. This study contains very important results and I believe that the revised manuscript is in great shape and can be published as is.

Reviewer #1:

Comment 1: The main contribution of this manuscript is the realization of $n>2$ photon dependencies by sequentially actuating in a stepwise manner (two photons at a time) on the molecular activation and photoswitching properties. This is based on a modular scheme where a two-photon-active fluorophore is deactivated by FRET by a secondary chromophore which can be transformed into its negatively photochromic state (which is not energy accepting). Once this secondary chromophore is switched, the energy transfer channel is deactivated, thereby “allowing” the two-photon chromophore to have a high emission yield. A critical issue is a low-concentration of the fluorescent state which in turn depends on the rate of re-conversion from the photoswitched state to the initial state. This is an interesting innovative scheme. I recommend publication after minimal issues are considered by the authors.

Response 1: We thank the reviewer for the very positive feedback on our work. Please find below a point-by-point answer to all comments and suggestions.

Minor issues

Comment 2: Should comment on the two photon cross sections of the PS themselves, in some cases, the PAP systems show relatively low tpa cross sections (of less than 20 GM). It is possible that from the conjugation of the PS, they themselves, could have competing two-photon absorption (this could be a positive characteristic as the objective of the switching may not require that the two-photons are absorbed by the PAP systems, but by the PS themselves).

Response 2: Indeed, the reviewer is correct. Any direct absorption of the 800 nm excitation light by the PS themselves (which we cannot exclude with absolute certainty) will not compromise the function. This is due to the fact that it does not matter how the PS units reach the excited state (FRET sensitized or direct excitation). We have added a sentence on page 6 to emphasize this observation:

Direct excitation of PS_C , if any, will not compromise the 4for2 function as it would trigger the same isomerization process to yield the desired PS_{CL} form.^{29, 30}

While no experimental data are available for the 1,1'-binaphthyl-bridged imidazole dimers, we have added a reference to a previous study reporting on the negligible two-photon cross section of similar DASA derivatives (ref. 29: 10.1039/d3sc01223a). Please note that some of the reference numbers changed due to this modification.

Comment 3: The authors should include experiments where (possibly at low temperatures where the thermal back isomerization is inhibited) it is shown which is the increase in the steady state emission from the 2PAS as the system evolves into a photo-stationary state. This would directly indicate which is the amount of fluorescence present in the system before the photo-transformation takes place, showing the dynamic range for the emission-turn-on. Although this is presented in terms of emission lifetimes, the steady state emission increase at the focal point of the two-photon system would be interesting.

Response 3: The suggested experiment is highly relevant for this work. Any attempts to perform the experiment, however, were not successful because the excitation intensities required for complete isomerization to the PS_{CL} -2PAP isomer triggered substantial photodegradation. Low temperature measurements would not be straightforward as these require substantial changes to the configuration of the setup (adding a cryostat and set the

sample in a special cell to stand the low temperatures). Furthermore, comparisons would be difficult, as the beams cross several windows, the pulse duration changes, etc.

However, an alternative measure of the indicated dynamic range is to compare the steady-state emission intensity of the dyads/triads in the PS_C-2PAP form to the corresponding steady-state emission intensity for the respective 2PAP monomer, using optically matched samples with respect to 2PAP absorption. The ratio between the intensity of the dyads/triads and the respective 2PAP monomer is referred to as the background emission. We have determined the background emission for all compounds, and we present the results in a new table in the Supporting Information (current Table S8; numbering of tables at the Supporting Information has been changed accordingly). The dynamic range is the inverse of the background emission.

Table S8. Background emission of all compounds under study.

Compound	Background emission ^a
DASA/FRT	< 0.12
DASA/Napht	< 0.14
NOMe/Napht	< 0.04
asyNOMe/Napht	< 0.03
NOMe/Phtha	< 0.03
NTPA/Napht	< 0.01

^a The background emission is determined as the ratio between the fluorescence intensities of the PS_C-2PAP dyad/triad and the corresponding 2PAP monomer. As the excitation light for emission readout triggers isomerization to the fluorescent form, we have given the background emission as an upper limit.

Comment 4: The authors should consider giving more molecular information right in the abstract of the molecules which are the actuators in these designs. The general concepts are well described, but to several areas, more rapid info about which are the actual molecular entities and how their intrinsic properties add to their combined value would be of great value in the abstract itself.

Response 4: We thank the reviewer for this constructive suggestion. We have rewritten the abstract, and included the following phrasing (page 2):

In these designs, two-photon absorbing push-pull fluorophores that function as FRET-donors are linked to T-type negative PS FRET-acceptors, e.g., donor-acceptor Stenhouse adducts (DASA) or 1,1'-binaphthyl-bridged imidazole dimers. FRET-sensitized isomerization of PS is delicately balanced by reverse thermal isomerization and results in non-linearly potentiated fluorescence with a quartic response upon two-photon excitation, implying superior spatial resolution potential.

Comment 5: It is not uncommon that non-linear excitation schemes show dependencies that are larger than square due to 3PA, etc. Would it be possible to show that in same concentrations with the same 2PAPS, the power dependence turns from square to quartic (for the dyads) in comparison?

Response 5: Prior to all relevant experiments, calibration of our setup was performed using the 2PA standard Rhodamine 6G. The resulting log-log plots yielded a slope near 2 which ensures that we are indeed working in the 2PA regime. A typical example is shown below.

Comment 6: The general concepts related to a sequential set of effects (each with 2P dependence) is clever and of significant potential value. I think the authors should also be explicit about the challenges that this kind of implementation will face. Some comments about this will help future researchers about additive contributions in this line. For example, which are the challenges in terms of solubility and how the photochromic and thermal dynamics will be affected by the solvent (for example, in DASAs this is highly affected by the solvent). Different negative-photoswitches will have quite different behaviors in different biological environments.

Response 6: This is a very valid point. As the reviewer is pointing out, shifting from a non-polar organic solvent to water (or even biologically more relevant environments) will be challenging due to the reasons indicated. We have added a short outlook-style text to the conclusion section on page 15:

The transition to biologically more relevant environments will require the careful design of molecular systems that meet the application-specific criteria (water solubility, polarity dependent photochemical/photophysical properties etc.). Nonetheless, the 4for2 approach showcases...

Reviewer #2:

Comment 1: This manuscript presents a very interesting study about the development of chemically linked dyads and tryads composed of 2PA fluorophores and T-type photoswitches. The experiments were well performed and the results are convincing. A lot of data is presented and discussed with an overall demonstration of quartic excitation dependence for 6 different dyads and tryads composed of 3 different 2PA fluorophores and 3 different T-type photoswitches (why were those 6 selected out of the 9 (or more when considering the dyads and tryads) possible combinations?). This is very impressive work that significantly extends beyond the authors' previous work using a dyad with a 2PA fluorophore and a P-type photoswitch. The proof-of-principle demonstration has the potential to open many powerful experimental possibilities for imaging with high spatial resolution and therefore, this manuscript is well suited for Nature Communications. I recommend the publication of this manuscript after revision (see comments below) to make the results easier to understand and to clarify some parts of the study that may not be clear to everyone (including me...).

Response 1: We thank the reviewer for the very positive feedback on our work.

In relation to the reviewer's question about the selection of six compounds out of several other possible combinations, our selection aimed to combine (i) FRET efficiency (we only considered the most optimal D-A combinations with respect to the seminal FRET reaction) with (ii) synthetic feasibility (the complex preparation procedures required for the synthesis of these compounds preclude certain possible combinations).

Please find below a point-by-point answer to all other comments and suggestions.

Comments:

Comment 2: I would recommend to clearly mention the differences compared to the authors' previous work with the P-type PS dyads. Considering the broad readership of Nat Comm, the differences (which are significant) may not be obvious to everyone.

Response 2: We agree fully with the suggestion to contrast our present and previous work (to emphasize the effect of including a T-type negative photoswitch rather than a P-type). We have added a sentence on page 6:

An illustrating example of this saturation effect is a previous work of ours where a P-type diarylethene was used as the photoswitch.²²

Comment 3: Figure 3 should also show the absorption of NTPA(CL), such that the missing (or negligible) spectral overlap can be appreciated. It could be added as a graph c (with same wavelength range) below the other two to avoid too many spectra in one graph.

Response 3: We have performed experiments to obtain the UV/vis absorption spectrum of NTPA_{CL}. Due to the establishment of a photothermal stationary state (even at lower temperatures), the spectrum of pure NTPA_{CL} was obtained by deconvolution analysis of the spectrum of an incompletely isomerized sample. The spectrum of NTPA_{CL} is now incorporated into Figure 3a (shown below). The negligible absorption at wavelengths longer than 450 nm clearly hints on an insignificant spectral overlap with the emission of Napht, which in turn indicates the absence of FRET between NTPA_{CL} and Napht.

Figure 3. Absorption and emission spectra of model compounds **Napht** and **NTPA** vs. triad **NTPA/Napht**. (a) Absorption (filled areas) and emission (solid line) spectra in toluene of model compounds **NTPA_C** (blue), **NTPA_{CL}** (purple), and **Napht** (black). Due to the efficient thermal isomerization to yield **NTPA_C**, we could not completely convert the sample to the **NTPA_{CL}** isomer. The absorption spectrum of **NTPA_{CL}** was therefore obtained by deconvolution analysis.

Comment 4: P11: The authors suggest that the spectra show an electronically decoupled system. However, it seems that the blue part of the composed spectrum is higher (comparing for example the intensities at 300 and 600 nm for both absorption spectra) than would be expected from a 2:1 linear combination in the triad system. Are there maybe free Napht in the system or some parts of NTPA in the triad are already in the colorless form (which would show stronger blue absorption – another reason to show that spectrum in Figure 3)?

Response 4: We thank the reviewer for this keen observation. We have reviewed, on our initiative, the absorption spectra of all dyads and triads by comparing them with the linear combination of the absorption of their respective components. We observed deviations at shorter wavelengths only for NTPA/Napht (see below) and NOME/Phtha, and have recorded new spectra for these compounds at lower concentrations. These new spectra align much better with those obtained from linear combination, and are incorporated in Figure 3 and Figure S5 for NTPA/Napht and NOME/Phtha, respectively.

We ascribe the deviations at shorter wavelengths to scattering due to the formation of aggregates (the molecular size of the compounds is rather considerable), as the use of lower concentrations dramatically improved the agreement between experimental spectrum and linear combination.

Comment 5: P12: How was the Förster distance of 4.82 nm calculated? I also do not think that a 3-digit accuracy is realistic. Please change to 4.8 nm and provide an error estimation. Based on overlaps and uncertainties with spectra in different conditions and solvents, I would expect 4.8 +/- 0.5 nm or alike. But maybe I am wrong...

Response 5: We agree with the reviewer that the use of three significant digits in the estimated FRET efficiencies is too precise. We have taken the advice of the reviewer and changed to two significant digits and considered uncertainties in the measurements ($48 \pm 2 \text{ \AA}$ in this case). The details of how we used the FRET formalism to estimate the efficiencies are described in the Supporting Information Section 3 (Theoretical calculations).

Comment 6: It seems that the FRET analysis was only performed with time-resolved spectroscopy (comparing lifetimes of D and DA). What about the fluorescence spectra of D and DA? They are mentioned but not shown (or maybe they are somewhere in the 100 pages SI). I assume that one cannot see much (if the quenching is really close to 100%) but it would be nice to have the spectra next to the lifetime data (in the SI section 4) and also analyze the intensities for FRET efficiency determination.

Response 6: We agree with the reviewer that comparing the steady-state emission intensities of D (i.e., 2PAP) and D-A (i.e., PS_c-2PAP) would in principle allow for an estimation of the FRET efficiencies. However, from the corresponding time-resolved measurements (supported also by modelling using the FRET formalism) we know that the FRET efficiencies are very close to unity. This implies, as the reviewer points out, that the steady-state emission from D-A is expected to be extremely weak (corresponding to fluorescence quantum yields on the order of 10^{-4}). This is why a 1% impurity of the unquenched D monomer (in the D-A sample) would give rise to 10 times stronger emission compared to the emission from D-A. The same is true if the thermal equilibrium of D-A contains 1% of the fluorescent isomeric form. We know that this so-called background emission (see point 3, reviewer 1) could be as high as 14%. This is why we would substantially underestimate the FRET efficiencies if we based these numbers on steady-state data. To illustrate this, we included information on the background emission for each compound in Table S8 in the Supporting Information (see also response 3 to reviewer 1).

Comment 7: Validation: Wait a minute... Would two entangled 2PA processes not lead to a slope of 16 when there is quartic dependence (2^4)? The slope of 4 shows a quadratic dependence (2^2), which would mean the authors showed 2 for 2 and not 4 for 2? Or is that the main difference compared to their previous dyad? Would that not also mean that the previous dyad (which had a slope of 10) would be much better when it comes to spatial resolution?

Response 7: It is very tempting to arrive at the formulated conclusion. However, two entangled 2PA processes will not result in a slope of 16, nor is the slope of the log-log-plot of our previous dyad 10. The number of 16 simply comes from the fact that when doubling the excitation intensity in a 4P (quartic) process, the fluorescence intensity is expected to increase by a factor of 16 (2^4). In the same way if we triple the excitation intensity we would arrive to a factor of 3^4 (81). The slope of the log-log-plot, however, is expected to be 4 for a process that depends on a quantity with the power of 4.

Comment 8: Conclusion: “In principle, one could also imagine P-type PS for the same purpose...”. But that was already shown in their previous dyad, right? And no two-color excitation was necessary.

Response 8: In the previous dyad, we only observe an enhanced non-linearity (maximum emission enhancement of 10 for doubling the excitation intensity) during a limited time period after exposure to the excitation light. Upon extended irradiation, we isomerize 100% of the previous dyad to the fluorescent isomer. Hence, we are left with a conventional two-photon behavior, as the concentration of the fluorescent isomer no longer increases with exposure time. This is referred to as saturation, and we have explained the effect of saturation in the manuscript on page 6. With the previous dyad, we did not attempt to overcome the saturation problem by using two-color excitation. To emphasize the effect of saturation and also to differentiate in greater detail the difference between the previous dyad and the compounds reported in the present manuscript, we have added a sentence on page 6:

An illustrating example of this saturation effect is a previous work of ours where a P-type diarylethene was used as the photoswitch.²²

Reviewer #3

Comment 1: This study reports a clever design of fluorescent dyes for "four-photon-like" absorption. For this purpose, the authors coupled fluorophores to photoswitches that show a rather fast thermal relaxation upon photoisomerization. The stable colored isomer of the photoswitch quenches the fluorescence emission, while the metastable non-colored isomer does not. In this way, a two photon excitation leads to a quartic dependence of the emission intensity on the excitation intensity.

Since the paper contains experiments on a set of dyes combining different photoswitches and fluorophores, it is likely that the concept can be generalized to other dyes with matching spectra and fast relaxation of the photoswitch.

This paper is highly relevant since the concept described here can be used to enhance the spatial resolution of optical microscopy without unwanted side effects such as irreversible saturation.

The experimental work is rigorous and well-described. Obviously the impact of the paper would be significantly increased if the authors included a proof of concept of a microscopy experiment, however in my view the current manuscript with its focus on molecular design and spectroscopy merits publication.

Response 1: We thank the reviewer for this very positive assessment of our work. We agree that the experimental demonstration of improved spatial resolution would add to the impact. However, it goes beyond this proof-of-concept study. We plan to address this feature in future research activities in our laboratories.